# Long-term outcomes of ivermectin-albendazole *versus* albendazole alone against soil-transmitted helminths: Results from randomized controlled trials in Lao PDR and Pemba Island, Tanzania

**Ladina Keller**[1,2], **Sophie Welsche**[1,2], **Chandni Patel**[1,2], **Somphou Sayasone**[1,2,3], **Said M. Ali**[4], **Shaali M. Ame**[4], **Jan Hattendorf**[1,2], **Eveline Hürlimann**[1,2], **Jennifer Keiser**[1,2]*

**1** Medical Parasitology and Infection Biology, Helminth Drug Development Unit, Swiss Tropical and Public Health Institute, Basel, Switzerland, **2** University of Basel, Basel, Switzerland, **3** Department of International Program for Health in the Tropics, Lao Tropical and Public Health Institute, Vientiane, Lao People's Democratic Republic, **4** Public Health Laboratory Ivo de Carneri, Chake Chake, Pemba, Zanzibar, Tanzania

* jennifer.keiser@swisstph.ch

## Abstract

### Background

Preventive chemotherapy is the cornerstone of soil-transmitted helminth (STH) control. Long-term outcomes and adequate treatment frequency of the recently recommended albendazole-ivermectin have not been studied to date.

### Methodology/principal findings

Double-blind randomized controlled trials were conducted in Lao PDR, Pemba Island, Tanzania and Côte d'Ivoire between 2018 and 2020 to evaluate the efficacy and safety of ivermectin-albendazole *versus* albendazole-placebo in *Trichuris trichiura*-infected individuals aged 6 to 60. In the framework of this study, in Lao PDR 466 and 413 participants and on Pemba Island, 558 and 515 participants were followed-up six and 12 months post-treatment, respectively. From each participant at least one stool sample was processed for Kato-Katz diagnosis and cure rates (CRs), egg reduction rates (ERRs) and apparent reinfection rates were calculated. If found helminth-positive at six months, participants were re-treated according to their allocated treatment.

Long-term outcomes against *T. trichiura* based on CRs and ERRs of ivermectin-albendazole compared to albendazole were significantly higher at six months in Lao PDR (CR, 65.8 *vs* 13.4%, difference; 52.4; 95% CI 45.0–60.0; ERRs, 99.0 *vs* 79.6, difference 19.4; 95% CI 14.4–24.4) and Pemba Island (CR, 17.8 *vs* 1.4%, difference; 16.4; 95% CI 11.6–21.0; ERRs, 84.9 *vs* 21.2, difference 63.8; 95% CI 50.6–76.9) and also at 12 months in Lao PDR (CR, 74.0 *vs* 23.4%, difference; 50.6; 95% CI 42.6–61.0; ERRs, 99.6 *vs* 91.3, difference 8.3; 95% CI 5.7–10.8) and Pemba Island (CR, 19.5 *vs* 3.4%, difference; 16.1; 95% CI 10.7–21.5; ERRs, 92.9 *vs* 53.6, difference 39.3; 95% CI 31.2–47.4) respectively.

**Data Availability Statement:** All relevant data are within the manuscript and its Supporting information files.

**Funding:** JK is grateful to the Bill and Melinda Gates Foundation for financial support (OPP1153928) The funders had no role in study design, data collection and analysis, decision to publish, or preparation of the manuscript.

**Competing interests:** The authors have declared that no competing interests exist.

**Abbreviations:** CI, confidence interval; CR, cure rates; EPG, eggs per gram; ERR, egg-reduction rates; GM, geometric mean; MDA, mass drug administration; OD, odds ratio; PC, preventive chemotherapy; RCT, randomized controlled trial; SAC, school-aged children; SES, socio-economic status; STH, soil-transmitted helminth; WHO, World Health Organization.

Apparent reinfection rates with *T. trichiura* were considerably higher on Pemba Island (100.0%, 95% CI, 29.2–100.0) than in Lao PDR (10.0%, 95% CI, 0.2–44.5) at 12 months post-treatment for participants treated with albendazole alone.

## Conclusions/significance

The long-term outcomes against *T. trichiura* of ivermectin-albendazole are superior to albendazole in terms of CRs and ERRs and in reducing infection intensities. Our results will help to guide decisions on how to best use ivermectin-albendazole in the context of large-scale PC programs tailored to the local context to sustainably control STH infections.

## Trial registration

ClinicalTrials.gov registered with clinicaltrials.gov, reference: NCT03527732, date assigned: 17 May 2018.

### Author summary

Around 1.5 billion people are infected with the gastrointestinal dwelling nematodes, the so-called soil-transmitted helminths (STHs). Especially chronic high-intensity infections with these parasites can cause substantial morbidity in endemic regions. Preventive chemotherapy, which is the regular administration of a single dose of an anthelmintic drug to at-risk populations, aims to reduce morbidity by lowering the prevalence and intensity of STH infections. Due to the suboptimal efficacy of these recommended single dose monotherapies, particularly against *Trichuris trichiura*, the use of drug combinations with dissimilar modes of action may enhance treatment efficacy. In double-blind randomized controlled trials in Lao PDR and on Pemba Island, Tanzania, we examined the long-term outcomes of ivermectin-albendazole *versus* albendazole alone, against *T. trichiura*. We assessed the infection status, apparent reinfections, new infections and the change of infection intensity for the three major STH species six and 12 months post-treatment. The long-term outcomes (in terms of cure and egg-reduction rates) of the ivermectin-albendazole combination therapy against *T. trichiura* were significantly higher than that of albendazole alone at all-time points and in both countries. Bi-annual treatment intervals using the ivermectin-albendazole combination might be necessary to sustainably decrease transmission of STH infections.

## Background

Infection with soil-transmitted helminths (STHs) (i.e. *Ascaris lumbricoides*, *Trichuris trichiura* and hookworms) is the most common and widespread parasitic human disease worldwide, mostly affecting marginalized populations in tropical and subtropical regions with limited access to adequate water, sanitation and hygiene [1–4]. STHs live in the intestine and are passed in the feces of an infected human host. Infections occur as a result of contact with infective stages in contaminated water sources, food or soil [5]. Approximately 1.5 billion individuals are infected with STHs worldwide, accounting for an estimated global disease burden of 1.92 million disability-adjusted life years in 2017 [6]. Diseases accompanying these infections cause substantial morbidity manifested as anemia [7], growth retardation and delayed

cognitive development in children [8], reduction in work performance in adulthood [9] and adverse pregnancy outcomes [10,11].

The mainstay of global STH control recommended by the World Health Organization (WHO) is preventive chemotherapy (PC) that is the periodic administration of an oral mono-dose of benzimidazoles carbamates (i.e. albendazole, mebendazole) to at-risk populations without prior diagnosis. These cost-effective deworming campaigns aim at reducing the worm burden to limit the progressive damage from chronic and recurrent infections [4,5]. Hence, the WHO goal for 2030 is morbidity control, defined as the reduction of moderate-to-heavy intensity infections to <2% in preschool-aged children (pre-SAC) and school-aged children (SAC) [12]. Yearly PC is recommended as a public health intervention for all children above 12 months of age in areas where any STH prevalence is between 20% and 50%, bi-annual administration is recommended where the prevalence exceeds 50% [13]. In settings where morbidity control has been achieved but the risk of transmission continues, a progressive reduction in PC frequency rather than stopping treatment programs is recommended to keep morbidity levels low [12].

While PC may be useful to control morbidity, it is less effective in transmission control [14–16], as the environmental contamination with eggs and larvae and individuals' exposures to environmental contamination remain high [17]. Hence, individuals may acquire infections rapidly after mass drug administration (MDA) campaigns, especially in the absence of adequate water supply, sanitation and hygiene [1,8,18].

The use of drug combinations with dissimilar modes of action might enhance treatment efficacy, as the recommended single dose monotherapies show limited efficacy, particularly against *T. trichiura* [19]. The combined use of ivermectin-albendazole has been recently included in the WHO Model List of Essential Medicine for treating intestinal helminths [20].

In a clinical trial conducted between 2018 and 2020 in Côte d'Ivoire, Lao People's Democratic Republic (PDR) and Pemba Island, Tanzania, we examined the efficacy of ivermectin-albendazole *versus* albendazole alone. In brief, the ivermectin-albendazole combination revealed significantly higher cure rates (CR, 65.7% *vs* 8.2% for Lao PDR, and 48.6% *vs* 6.1% for Pemba Island) and higher egg-reduction rates (ERR, 99.2% *vs* 68.8% in Lao PDR and 98.3% *vs* 57.1%) on Pemba Island against *T. trichiura* than albendazole alone. However, in Côte d'Ivoire, albendazole alone and also in combination with ivermectin showed low efficacy in terms of both CR (10.2% *vs* 13.8%), and ERR (64.1% *vs* 70.2%). Detailed short-term efficacy results (14–21 days post-treatment) are published elsewhere [21].

The aim of the present study was to evaluate long-term outcomes (CR, ERR) and reinfection patterns of ivermectin-albendazole combination compared to albendazole alone six and 12 months post-treatment (including a re-treatment at six months in helminth positives) against *T. trichiura* and concomitant STH infections.

## Methods

### Ethics statement

Prior to the study initiation, ethical clearance was granted by the Ethics Committee of Northwestern and Central Switzerland (EKNZ; reference no: BASEC Nr Req-2018-00494), the National Ethics Committee for Health Research, Ministry of Health in Lao PDR (reference no: 093/NECHR) and the Zanzibar Medical Research and Ethics Committee (ZAMREC, reference no.: ZAMREC/0003/Feb/2018) and the Comité National d'Éthique et de la Recherche, Ministère de la Santé et de Lutte contre le SIDA (reference no.: 088–18/MSHP/CNESVS-km) and the Direction de la Pharmacie, du Médicament et des Laboratoires (reference no. ECCI00918) in Côte d'Ivoire. Trial details are summarized in the published trial protocol [22] and in the

trial registration (clinicaltrials.gov, reference: NCT03527732, date assigned: 17 May 2018). Written informed consent was sought from adult participants and caregivers of children aged six to 17 prior to study enrolment. Children aged below the age of adulthood gave oral assent.

## Study participants and design

The presented data derived from a double-blind, standard of care-controlled randomized controlled trial (RCT) conducted in the Nam Bak district in Luang Prabang Province in Lao PDR and in the Chake Chake district, South Pemba on Pemba Island, Tanzania between September 2018 to July 2020. Data from the third study site, Côte d'Ivoire, are not presented here as the study was stopped after the first follow-up time point.

Consenting participants were eligible for the trial if they had provided two fecal samples at baseline, were aged between six and 60 years, weighed at least 15 kg, were positive for *T. trichiura* infection in at least two slides of the quadruple Kato-Katz smears and had an infection intensity of at least 100 eggs per gram (EPG) of stool. After an initial clinical examination, participants were excluded if they were pregnant or lactating in the 1st week after birth, had a major systemic illness, had clinical malaria, had a history of severe acute or chronic diseases, or met other exclusion criteria listed in the trial protocol [22].

The clinical trial involved four assessment time points: baseline, 14–21 days, six and 12 months post-treatment. Participants positive for any STH were re-treated according to their study arm at six months. Treated participants were asked to provide two stool samples at each time point, whereof some provided no follow-up data, some incomplete and some complete follow-up data.

## Randomization and masking

The trial statistician, who was not involved in any field work, provided a computer-generated stratified (baseline infection intensity: light infections (<1000 EPG), and moderate/heavy infections (≥1000 EPG)) allocation sequence with variable block size (blocks of four, six or eight). Eligible participants were allocated 1:1 to either albendazole (400 mg) plus ivermectin (200 μg/kg body weight), or albendazole (400 mg) plus placebo. Matching ivermectin placebo tablets were provided by the University of Basel. Treatment allocation was concealed: prior to trial initiation two people independent to the study prepared sealed, opaque, and sequentially numbered envelopes containing the treatment arm for each participant. Participants, investigators, outcome assessors and the trial statistician were blinded during the whole study period.

## Study procedures

A census was conducted at the start of screening, during which the name, sex, age and village was recorded and a unique pseudonymized identifier was assigned to all individuals. A questionnaire was applied to an adult from each household by a trained interviewer in the local language (Lao or Swahili) during the trial to assess the socio-economic status (e.g. possession of a number of household items).

Consenting participants were asked to provide two fresh morning stool samples, preferably on consecutive days and within a maximum of 5 days apart at each time point assessment. Collected stool samples were kept in a cooling box containing ice packs while being transported to the laboratory. Samples were examined with quadruplicate Kato-Katz microscopy within 24 h after collection for the detection of STH ova by experienced laboratory technicians following the WHO standard procedures [23]. The eggs of *T. trichiura*, *A. lumbricoides* and hookworm eggs were counted and recorded for each species separately. Ten percent of all Kato-Katz slides were randomly selected, re-labelled with new identification numbers and re-read for

*A. lumbricoides* and *T. trichiura* for quality control. To ensure quality of hookworm diagnosis, 10% of the stool samples were divided into two sub-samples and duplicate Kato-Katz were made from both containers and the findings compared. Additionally, in Lao PDR, stool samples were examined for *Opisthorchis viverrini* and *Strongyloides stercoralis*. Egg counts of *O. viverrini* derived from the same Kato-Katz thick smears, whereas *S. stercoralis* samples were classified as larvae-positive or negative using the Baermann technique [24].

## Study outcomes

The outcomes reported here from this clinical trial were CR and ERRs of STH infections six and 12 months post-treatment, as well as apparent reinfection rates with *T. trichiura* and newly acquired infections with *A. lumbricoides* and hookworm at the two follow-up assessments.

## Sample size

For the sample size calculation we considered potential treatment efficacy of each treatment arm at study start and after re-treatment at 6 months in order to be able to identify a difference in the 12 month infection status between arms [22]. In brief, a CR of 30% using albendazole against *T. trichiura* and a CR of 50% using the ivermectin-albendazole combination therapy was assumed. Additionally, we assumed the same treatment efficacy and a reinfection risk of 10% six months post-treatment. Thus, 44% and 65% of participants treated with albendazole and ivermectin-albendazole were expected to be STH-negative at 12 months post-treatment respectively. Accounting for a loss to follow-up of 30% at six months and 40% at 12 months post-treatment (final assessment), we obtained a final sample size of 600 participants (300 per treatment arm) in each country, with a total of 1200 participants.

## Statistical analysis

Paper-based data were double-entered and cross-checked in EpiInfo version 3.5.4 (Centers for Disease Control and Prevention, Atlanta, United States of America) by two independent data clerks and merged into a single database for statistical analysis using StataIC15 (StatCorp.; College Station, TX). Figures were generated with R 3.4.0. The parasitological status of participants was described in terms of prevalence and infection intensity for *T. trichiura*, *A. lumbricoides* and hookworm and assessed for both treatment arms at baseline, 14–21 days (short-term), six and 12 months (long-term) post-treatment. In Lao PDR, the prevalence of co-infections with *O. viverrini* and *S. stercoralis* at each time point was also taken into account. Results from the duplicate Kato-Katz thick smears from each of the two stool samples were summed and multiplied by a factor of six to be expressed as geometric mean (GM) EPG using the following formula: $(GM = \exp((\Sigma \log (EPG + 1))/n) - 1)$ [25]. Infection intensity was categorized as light (1–999 EPG), moderate (1000–9999 EPG) or heavy (>9999 EPG) for *T. trichiura*. Similarly, the intensity of *A. lumbricoides* and hookworm infections was classified according to WHO recommendations, based on guidelines established by Montresor *et al.* [25].

## Long-term drug effect measures

CRs were defined as the proportion of egg-positive individuals who became egg-negative 14–21 days post-treatment. Long-term outcomes of the two treatments were measured in terms of CRs and extended ERRs. The CRs was defined by individuals testing negative at 14–21 days post-treatment as well as either at six and/or at 12 months post-treatment. The ERR, using GM

EPG, was calculated with the following formula: $ERR = \left(1 - \frac{e^{\frac{1}{n}\Sigma\log(EPG_{follow-up} + 1)} - 1}{e^{\frac{1}{n}\Sigma\log(EPG_{baseline} + 1)} - 1}\right)*100$. The melded binomial test with mid-p correction was used to calculate differences in CRs and its corresponding 95% CIs. Interval estimates for the differences in ERRs among the two treatment groups were estimated using Bootstrap resampling algorithms with 5000 replications. Odds ratios (ORs) for CRs with corresponding 95% CIs and the difference (%-points) among treatment arms in CRs were assessed using logistic regression. A crude logistic regression was used for the 6-months follow-up; while adjustment for participant's age and sex, socioeconomic status (SES), treatment arm, mean EPG at six months and co-infection of other parasitic infections (in Lao PDR only) were made for the 12-months follow-up. The SES was calculated using a household asset-based approach [26], where the weight of each household asset was determined using principal component analysis and an index score was generated. Households were classified into wealth quartiles according to their index score. Questions used to assess the SES are summarized in the supporting information (S1 Text).

## Reinfection measures

Apparent reinfection rates for all three major STH species were defined as individuals positive at baseline, negative at 14–21 days and positive at six and/or 12 months post-treatment. New infections were defined as individuals negative at baseline and 14–21 days post-treatment and positive at six or 12 months. As all participants, per study design, were positive for *T. trichiura* at baseline, new infections were only applicable for *A. lumbricoides*, hookworm and *S. stercoralis* (in Lao PDR only) infections. Due to the low number of cured participants for albendazole alone (n = 31/232) in Lao PDR and (n = 4/282) on Pemba Island 6 months post-treatment, the odds for being reinfected were not calculated. *P*-values <0.05 were considered statistically significant.

## Results

In total, 549 participants in Lao PDR and 613 participants on Pemba Island were randomized and treated with either (i) ivermectin-albendazole or (ii) albendazole-placebo at baseline (Fig 1). In Lao PDR 407, 466 and 413 participants provided at least one stool sample to assess infection status 14–21 days, six and 12 months post-treatment while on Pemba Island 581, 558 and 515 participated in the corresponding follow-up examinations. Of note, due to the COVID-19 pandemic, the sample collection at the 12 months follow-up in Lao PDR started with a delay of approximately six weeks.

### Baseline characteristics of country trial cohorts

Baseline parasitological and demographic characteristics are summarized in Table 1. 53.0% of participants in Lao PDR and 55.5% of participants on Pemba Island were female. The majority of participants (62.7%) in Lao PDR were adults (mean age of 26.8 years), while participants on Pemba Island were predominantly school-aged children (61.2%, mean age of 14.0 years). 321 (69.9%) of participants in Lao PDR were also infected with either *A. lumbricoides* or hookworm, and 195 (35.5%) were infected with all three STHs. On Pemba Island, 131 (21.4%) were co-infected with another STH and 44 (7.2%) were infected with all three parasites. In Lao PDR, 117 (21.3%) and 59 (10.8%) of the participants were also co-infected with *O. viverrini* and *S. stercoralis*, respectively (Table 1).

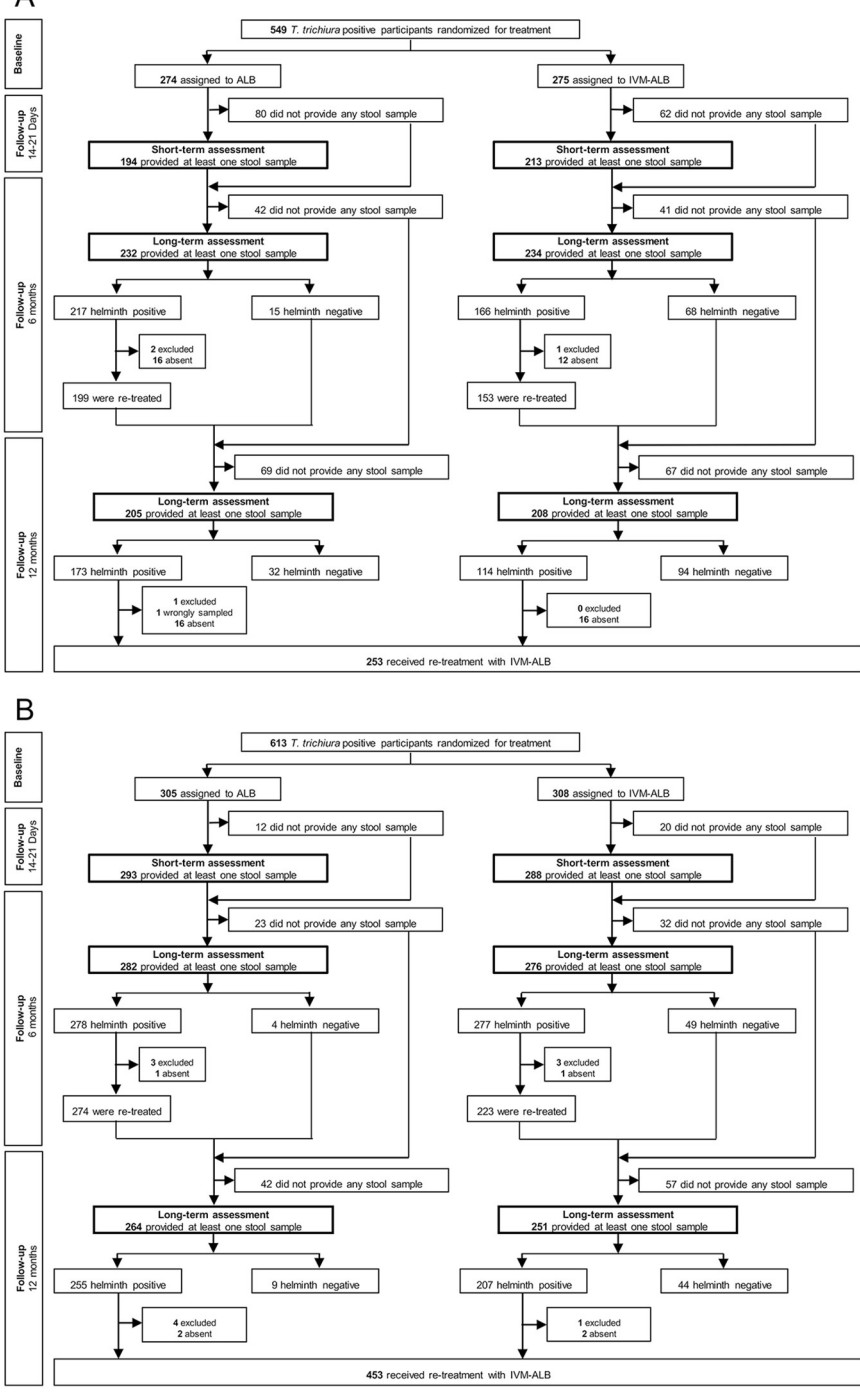

**Fig 1. Study design flow chart for Lao PDR (A) and Pemba Island (B).** Abbreviations: *T. trichiura*, *Trichuris trichiura*; ALB, albendazole monotherapy; IVM-ALB, ivermectin-albendazole combination therapy. In Lao PDR "helminth positive" includes infections with *Strongyloides stercoralis*

## Long-term outcomes against *T. trichiura* six and 12 months post-treatment

As participants were only enrolled if *T. trichiura* positive by design, prevalence of the study cohort was 100% at baseline (N = 549 in Lao PDR, N = 613 on Pemba Island). The short-term

**Table 1. Baseline characteristics of country trial cohorts.**

| Characteristics | Lao PDR | | Pemba Island | |
|---|---|---|---|---|
| | Albendazole monotherapy (n = 274) | Ivermectin-albendazole (n = 275)[a] | Albendazole monotherapy (n = 305) | Ivermectin-albendazole (n = 308)[b] |
| Age group, n (%) | | | | |
| School-aged (6–17 yrs) | 92 (33.6) | 113 (41.1) | 191 (62.6) | 184 (59.7) |
| Adults (18–60 yrs) | 182 (66.4) | 162 (58.9) | 114 (37.4) | 124 (40.3) |
| Sex, n (%) | | | | |
| Female | 144 (52.6) | 147 (53.4) | 171 (56.1) | 169 (54.9) |
| Male | 130 (47.4) | 128 (46.6) | 134 (43.9) | 139 (45.1) |
| *T. trichiura* infection | | | | |
| Geometric mean EPG | 369.8 | 361.0 | 462.8 | 463.2 |
| Infection intensity[c], n (%) | | | | |
| Light | 232 (84.7) | 232 (84.4) | 231 (75.7) | 234 (76.0) |
| Moderate | 42 (15.3) | 42 (15.3) | 74 (24.3) | 71 (23.1) |
| Heavy | 0 (0.0) | 1 (0.3) | 0 (0.0) | 3 (1.0) |
| *A. lumbricoides* infection | | | | |
| Infected, n (%) | 112 (40.9) | 96 (34.9) | 74 (24.3) | 90 (29.2) |
| Geometric mean EPG | 3991.2 | 3635.0 | 4515.3 | 2979.5 |
| Infection intensity[c], n (%) | | | | |
| Light | 58 (51.8) | 48 (50.0) | 35 (47.3) | 52 (57.8) |
| Moderate | 47 (42.0) | 44 (45.8) | 38 (51.4) | 37 (41.1) |
| Heavy | 7 (6.2) | 4 (4.2) | 1 (1.4) | 1 (1.1) |
| Hookworm infection | | | | |
| Infected, n (%) | 250 (91.2) | 253 (92.0) | 53 (17.4) | 42 (13.6) |
| Geometric mean EPG | 804.8 | 840.3 | 100.5 | 79.9 |
| Infection intensity[c], n (%) | | | | |
| Light | 182 (72.8) | 184 (72.7) | 53 (100.0) | 42 (100.0) |
| Moderate | 68 (27.2) | 69 (27.3) | 0 (0.0) | 0 (0.0) |
| Heavy | 0 (0.0) | 0 (0.0) | 0 (0.0) | 0 (0.0) |
| Total co-infections of *T. trichiura* | | | | |
| with 1 other STH | 159 (58.0) | 167 (60.7) | 85 (27.9) | 86 (27.9) |
| with 2 other STH | 104 (38.0) | 91 (33.1) | 21 (6.9) | 23 (7.5) |
| with *S. stercoralis*[d] | | | | |
| Infected/Surveyed, n (%) | 29/273 (10.6) | 30/274 (11.0) | ND | ND |
| with *O. viverrini*[e] | | | | |
| Infected, n (%) | 35 (12.8) | 33 (12.0) | ND | ND |

Abbreviations: *A. lumbricoides*, *Ascaris lumbricoides*; *T. trichiura*, *Trichuris trichiura*; *S. stercoralis*, *Strongyloides stercoralis*; *O. viverrini*, *Opisthorchis viverrini*; ND, not determined

[a] Two of the 275 participants were mistakenly randomized (age >60 yrs)

[b] One of the 308 participants was mistakenly randomized (<100 EPG)

[c] Infection intensities were classified according to WHO recommendations, based on guidelines established by Montresor *et al.* (1998) [25]

[d] Baermann technique to detect *S. stercoralis* infection was only applied in Lao PDR. Two Lao participants (one of each arm) have no *S. stercoralis* result.

[e] Detection of *O. viverrini* eggs was only applied in Lao PDR

efficacies 14–21 days post-treatment are summarized in Table 2 and presented in more detail elsewhere [21].

In both settings, CRs on *T. trichiura* six months post-treatment were significantly higher for ivermectin-albendazole compared to albendazole alone [Lao PDR: 65.8% *vs* 13.4%;

**Table 2. Short and long-term outcomes and reinfection data for ivermectin-albendazole and albendazole alone against *T. trichiura* in Lao PDR and Pemba Island.**

| | Time point | Albendazole alone | Ivermectin-albendazole | Difference (%-points) | 95% CI | Odds Ratio for being cured[abc] (95% CI) |
|---|---|---|---|---|---|---|
| **Lao PDR** | | | | | | |
| N participants randomized (%) | Baseline | 274 (100) | 275 (100) | | | |
| N participants cured/N surveyed (CR, %) | 14–21 days | 16/194 (8.2) | 140/213 (65.7) | 57.5 | [50.0–64.9]* | |
| N participants negative (CR, %) | 6 months | 31/232 (13.4) | 154/234 (65.8) | 52.4 | [45.0–60.0]* | 12.5 [7.8–19.9] |
| | 12 months | 48/205 (23.4) | 154/208 (74.0) | 50.6 | [42.6–61.0]* | 9.9 [3.9–25.4] |
| EPG geometric mean | Baseline | 369.8 | 361 | | | |
| | 14–21 days | 115.5 | 3.0 | | | |
| | 6 months | 77.3 | 3.6 | | | |
| | 12 months | 33.5 | 1.7 | | | |
| ERR geometric | 14–21 days | 68.8 | 99.2 | 30.4 | [23.6–37.2]* | |
| Extended ERR | 6 months | 79.6 | 99.0 | 19.4 | [14.4–24.4]* | |
| | 12 months | 91.3 | 99.6 | 8.3 | [5.7–10.8]* | |
| N re-treated participants (%) | 6 months | 199 (72.6) | 153 (55.6) | | | |
| Apparent reinfections (%) | 6 months | 2/15 (13.3) | 18/126 (14.3) | -1.0 | [-19.2–17.3] | |
| N pos/N cured[d] (%) | 12 months | 1/10 (10.0) | 13/93 (14.0) | -4.0 | [-23.8–15.9] | |
| **Pemba Island** | | | | | | |
| N participants randomized (%) | Baseline | 305 (100) | 308 (100) | | | |
| N participants cured/N surveyed (CR, %) | 14–21 days | 18/293 (6.1) | 140/288 (48.6) | 42.5 | [36.1–48.9]* | |
| N participants negative (CR, %) | 6 months | 4/282 (1.4) | 49/276 (17.8) | 16.4 | [11.6–21.0]* | 15.0 [5.3–42.2] |
| | 12 months | 9/264 (3.4) | 49/251 (19.5) | 16.1 | [10.7–21.5]* | 4.7 [2.1–10.5] |
| EPG geometric mean | Baseline | 462.8 | 463.2 | | | |
| | 14–21 days | 198.5 | 8.0 | | | |
| | 6 months | 356.4 | 70.3 | | | |
| | 12 months | 220 | 33.8 | | | |
| ERR geometric | 14–21 days | 57.1 | 98.3 | 41.2 | [33.0–49.3]* | |
| Extended ERR | 6 months | 21.2 | 84.9 | 63.8 | [50.6–76.9]* | |
| | 12 months | 53.6 | 92.9 | 39.3 | [31.2–47.4]* | |
| N re-treated participants (%) | 6 months | 274 (97.1) | 222 (80.4) | | | |
| Apparent reinfections (%) | 6 months | 15/18 (83.3) | 88/130 (67.7) | 15.6 | [-3.3–34.6] | |
| N pos/N cured[d] (%) | 12 months | 3/3 (100) | 25/42 (59.5) | 40.5 | [25.6–55.3] | |

Abbreviations: CI, confidence interval; CR, cure rate; EPG, egg per gram of stool; ERR, egg reduction rate; ND, not determined

Participants were re-treated if found positive for *T. trichiura*, *A. lumbricoides*, hookworm or *S. stercoralis* (in Lao PDR only)

Apparent reinfections were defined as participants positive at baseline, negative at 14–21 days and positive at six and/or 12 months post-treatment

[a] Albendazole-placebo is considered as the reference group

[b] Crude odds ratio for six months follow-up, adjusted odds ratio for 12 months follow-up

[c] Adjusted for age, sex, socio-economic status, treatment arm and mean EPG at 6 months

[d] N cured refers to all participants being cured at the time point assessment before; cured at 14–21 days assessment for the 6 months follow-up and cured at 6 months assessment for the 12 months follow-up, respectively

*Statistically significant (P-value <0.05)

difference 52.4%-points; 95% CI, 45.0–60.0; Pemba Island: 17.8% *vs* 1.4% [difference 16.4%-points; 95% CI, 11.6–21.0]. Similarly, the CRs on *T. trichiura* documented 12 months post-treatment were significantly higher for ivermectin-albendazole compared to albendazole alone in Lao PDR [difference 50.6%-points; 95% CI, 42.6–61.0] and also on Pemba Island [difference 16.1%-points; 95% CI, 10.7–21.5].

Geometric mean based ERRs following administration of ivermectin-albendazole were significantly higher than those obtained from albendazole alone in Lao PDR [difference 19.4; 95% CI, 14.4–24.4] and on Pemba Island [difference 63.8; 95% CI, 50.6–76.9] six months post-treatment. The geometric mean based ERR 12 months post-treatment were also significantly higher for ivermectin-albendazole in either setting (Table 2). According to the results of the univariate logistic regression analyses, the crude OR for being cured following treatment with the ivermectin-albendazole combination at 6 month post-treatment was 12.5 [95% CI, 7.8–19.9] in Lao PDR and 15.0 [95% CI, 5.3–42.2] on Pemba Island. The adjusted ORs for being cured at 12 months were 9.9 [95% CI, 3.9–25.4] in Lao PDR and 4.7 [95% CI, 2.1–10.5] on Pemba Island (Table 2).

The sample size for the apparent reinfection rates was limited, as only cured (at 14–21 days post-treatment) participants were included. Apparent reinfection rates were considerably higher on Pemba Island independent of the treatment. All participants receiving albendazole alone were reinfected at least once within the study period (apparent reinfection rate of 100%) on Pemba Island. Apparent reinfection rates in Lao PDR remained low at both assessment time points and also regardless the treatment arm. Of note, as CRs 14–21 days post-treatment with albendazole alone were low against *T. trichiura* (8.2% in Lao PDR and 6.1% on Pemba Island), the odds for apparent reinfections have not been calculated.

### *Trichuris trichiura* infection intensity dynamics

Moderately and highly infected participants were equally balanced among both treatment arms at baseline. In both study settings, most *T. trichiura* infections were of light intensity (Table 1). The ivermectin-albendazole combination led to a larger reduction of heavy and moderate *T. trichiura* infections into light infections, both at six and 12 months post-treatment, when compared to albendazole alone. In detail, baseline moderate to heavy infection intensities decreased from 15.3% (95% CI, 11.3–20.1) to 0.5% (95% CI, 0.1–2.6) in Lao PDR and from 24.0% (95% CI, 19.4–29.2) to 1.2% (95% CI, 0.3–4.3) on Pemba Island 12 months post treatment with ivermectin-albendazole. The dynamics of infection intensities of *T. trichiura* at baseline, 14–21 days, six and 12 months post-treatment for participants is depicted in Fig 2.

### Long-term drug outcomes against *Ascaris lumbricoides* six and 12 months post-treatment

Among participants included in the trial, 208 (37.8%) in Lao PDR and 164 (26.8%) on Pemba Island were co-infected with *A. lumbricoides* at baseline. Although treatment efficacies 14–21 days post-treatment were high among both treatment arms, follow-up prevalences at six and 12 months were similar to the pre-treatment levels among both treatments on Pemba Island, while slightly less participants were infected with *A. lumbricoides* six and 12 month following treatment with the ivermectin-albendazole combination in Lao PDR (summarized in S1 Table). While the ERRs at the first follow-up were high (ERR, 98.8–100), ERRs six months post-treatment were considerably lower (ERR, -5.7–82.2). In total, 43 (33.6%) of cured participants in Lao PDR and 62 (44.9%) on Pemba Island were found to be reinfected at the six months assessment time point. The majority of reinfected participants harbored a light

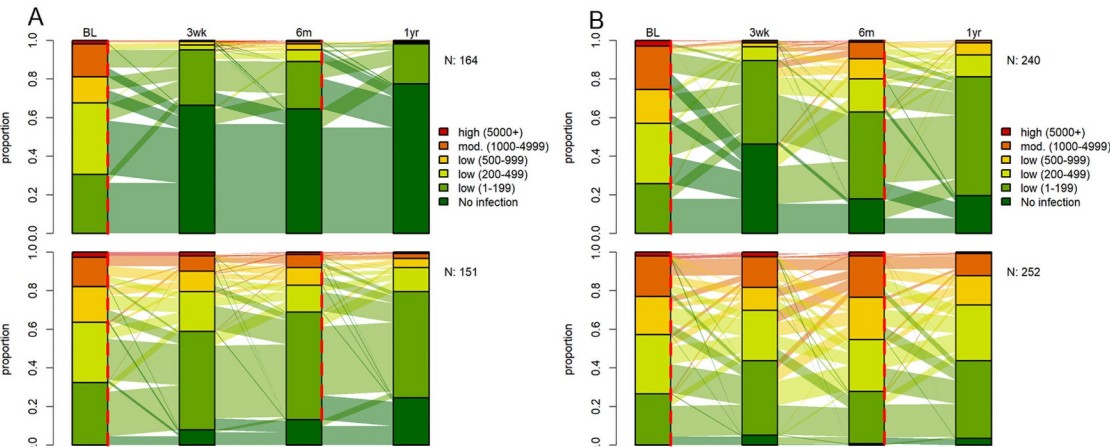

**Fig 2. Proportion of infection intensities of *T. trichiura* at baseline, 14–21 days, six and 12 months post-treatment in Lao PDR (A) and Pemba Island (B) for ivermectin-albendazole above and albendazole-placebo below.** 315 participants in Lao PDR and 492 participants on Pemba Island had a complete data set. Red dashed lines at baseline and follow up represent treatment time points. Abbreviations: BL, baseline assessment; N, number of participants with a complete data set; 3wk, 14–21 days post-treatment follow-up; 6m, 6 months post-treatment follow-up; 1yr, 12 months post-treatment follow-up.

*A. lumbricoides* infection. The number of new infections six and 12 months post-treatment was similar among both treatment arms. In total, new infections at 6 months post-treatment were observed for 28 (12.0%) participants in Lao PDR and for 63 (15.8%) participants on Pemba Island. Similar observations were obtained at the 12 months follow-up (S1 Table). The dynamics of infection intensities of *A. lumbricoides* at baseline, 14–21 days, six and 12 months post-treatment for participants is summarized in Fig 3.

## Long-term outcomes against hookworm six and 12 months post-treatment

At baseline, 503 (91.6%) participants in Lao PDR and 95 (15.5%) participants on Pemba Island were co-infected with hookworm. Among those infected with hookworm, 117 (25%) harbored

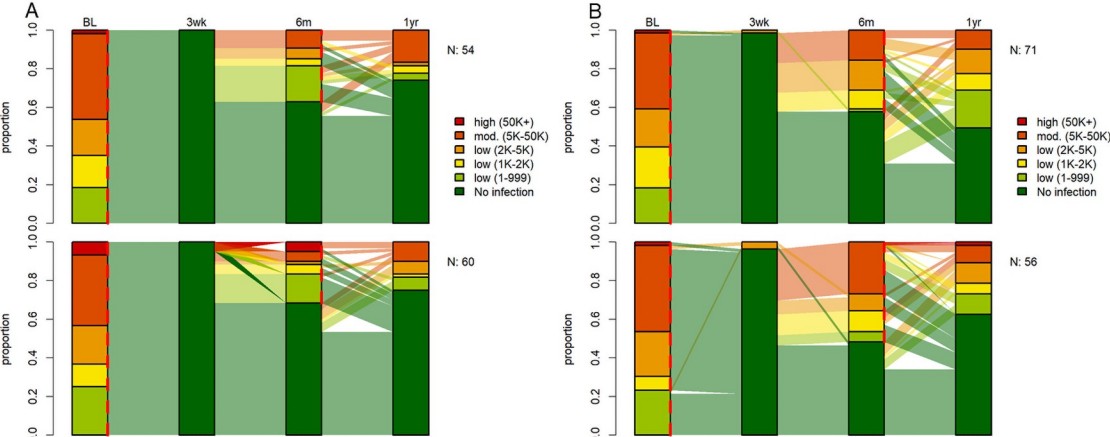

**Fig 3. Proportion of infection intensities of *A. lumbricoides* at baseline, 14–21 days, six and 12 months post-treatment in Lao PDR (A) and Pemba Island (B) for ivermectin-albendazole above and albendazole-placebo below.** 114 participants in Lao PDR and 127 participants on Pemba Island were initially infected with *A. lumbricoides* and had a complete data set. Red dashed lines at baseline and follow up represent treatment time points. Abbreviations: BL, baseline assessment; N, number of participants with a complete data set; 3wk, 14–21 days post-treatment follow-up; 6m, 6 months post-treatment follow-up; 1yr, 12 months post-treatment follow-up.

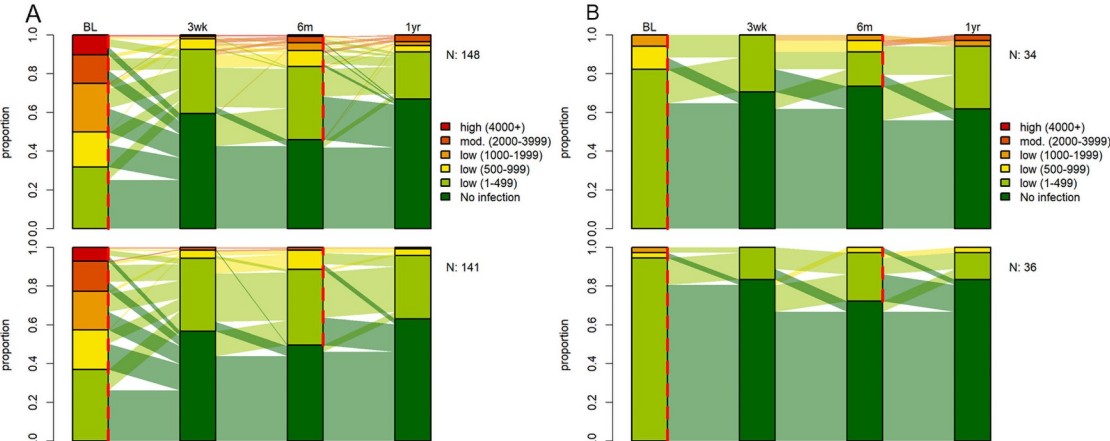

**Fig 4. Proportion of infection intensities of hookworm at baseline, 14–21 days, six and 12 months post-treatment in Lao PDR (A) and Pemba Island (B) for ivermectin-albendazole above and albendazole-placebo below.** 289 participants in Lao PDR and 40 participants on Pemba Island were initially infected with hookworm and had a complete data set. Red dashed lines at baseline and follow up represent treatment time points. Abbreviations: BL, baseline assessment; N, number of participants with a complete data set; 3wk, 14–21 days post-treatment follow-up; 6m, 6 months post-treatment follow-up; 1yr, 12 months post-treatment follow-up.

moderate infections in Lao PDR, while all hookworm infection were of light infection intensity on Pemba Island. Corresponding treatment efficacies at 14–21 days were moderate in Lao PDR and high on Pemba Island for both treatment arms (summarized in S1 Table) [21]. The CRs six and 12 months post-treatment remained moderate in Lao PDR and high on Pemba Island. In total, 238 (51.1%) participants in Lao PDR and 34 (5.8%) participants on Pemba Island were infected with hookworm six months post-treatment. Hookworm prevalence further dropped to 34.1% in Lao PDR at the 12 months assessment point, while it remained similar on Pemba Island. The hookworm infection intensity was markedly reduced in Lao PDR at the first follow-up assessment and remained at this level during the study period.

In total, 50 (26.2%) of cured participants in Lao PDR and 11 (18.0%) participants on Pemba Island were found to be reinfected at the six months assessment. In total, only 6 (20.7%) participants in Lao PDR and 9 (1.9%) participants on Pemba Island acquired a new hookworm infection at the six months assessment. The number of new infections 12 months post-treatment remained low among both treatment arms (S1 Table). The dynamics of infection intensities of hookworm at baseline, 14–21 days, six and 12 months post-treatment is presented in Fig 4.

## Discussion

The currently used standard drugs (i.e. albendazole and mebendazole) yield poor efficacy in clearing *T. trichiura* infections and reducing worm burden [27,28]. Additionally, the potential emergence of drug resistance is more topical than ever, yet only few other drugs have been tested for anthelmintic activity in humans or are currently in the pipeline [29]. Therefore, the development of new drugs, repurposing of available drugs or use of drug combinations to expand the armamentarium of treatment options for PC is of paramount importance. In the current study, we present detailed insights into the dynamics of ivermectin-albendazole, the most promising approved STH therapy to date using a bi-annual treatment schedule.

Most studies assessing *T. trichiura* prevalence post-treatment with the standard drugs, albendazole and mebendazole, report on the prevalence to regress to pre-treatment level

between six and 12 months [15,16,30,31], which is in complete agreement with our cohort receiving albendazole alone. The CRs of albendazole observed in our settings are even lower than CRs reported in a review and meta-analysis by Jia *et al*. [14]. Previous long-term studies struggled with low CR of the standard drug against *T. trichiura* and concise long-term conclusions on reinfection were lacking to date [14,32]. The results of our study show that the long-term outcomes (in terms of CRs and ERRs) of the ivermectin-albendazole combination therapy against *T. trichiura* was significantly higher than that of albendazole alone at all-time points and in both countries. It is worth highlighting that the excellent long term outcomes of ivermectin-albendazole are due to the combination's good short term efficacy, which are still present after one year of treatment, despite reinfection. A good long-term performance was particularly observed in Lao PDR, where the *T. trichiura* prevalence dropped from 100.0% to 23.0% within the study period of 12 months. Additionally, the combination therapy led to a larger reduction of moderate and heavy *T. trichiura* infections, the key indicator from a public health perspective [33], and successfully reduced the occurrence of these infections to below 1.5% within the 12 months in both countries. These results comply with the indicator for the 2030 target for STH elimination put forward by WHO [12].

On the other hand, the data derived from this study confirm that the current standard anthelminthic drug (albendazole alone) is not sufficiently effective and, thus, should only be used in combination with ivermectin or be replaced by more potent anthelminthics in future deworming programs in particular in settings with a high prevalence of *T. trichiura*. Of note, however, ivermectin-albendazole revealed an unsatisfactory short-term efficacy in Côte d'Ivoire that warrants further investigation and, hence, limits the generalizability of the presented long-term findings on ivermectin-albendazole.

Infection intensities usually recover slower than prevalences post-treatment [34], which might play an important role in the case of *T. trichiura* infections, as this parasite is known to be more persistent to treatment than other helminths [35]. Thus, the substantial reduction of EPG, as we have observed in the ivermectin-albendazole arm with a re-treatment over the 12 month period, is promising, even though the ivermectin-albendazole combination therapy did not seem to be efficacious in terms of CR in the case of Pemba Island. Shorter treatment intervals using the ivermectin-albendazole combination might be necessary to sustainably decrease the worm burden in persistent high-transmission settings such as Pemba Island, even though intervals of re-treatment between six months and 12 months are the most widely applied compromise between high impact and logistic feasibility [30]. If the aim is to interrupt STH transmission, several rounds of MDA dispensing ivermectin-albendazole might be also effective to reduce parasite populations to low levels and close to a breaking point, where recovery of the parasites may be insufficient to return back to baseline infection levels and thus, elimination of transmission could be reached [31].

Apparent reinfection rates with *T. trichiura* in Lao PDR at six and 12 months were considerably lower compared to Pemba Island, regardless of the treatment received. Even though open defecation practices might be a contributing factor to these high reinfection rates, it is unlikely to drive the discrepancy in our two study settings as self-reported open defaecation practices were high in both countries (S1 Fig). A likely explanation might be the high population density on Pemba as well as that STH and particularly *T. trichiura* prevalence on Pemba Island is considerably higher compared to Lao PDR, where we found a highly focal distribution of STH prevalence [36]. Consequently, the overall environmental contamination is presumably much higher on Pemba Island, leading to a higher risk of reinfection. Of note, *T. trichiura* eggs can remain viable and infective for several months [37], putting individuals at risk for reinfection from eggs persisting in the environment, without the need for the deposition of new infective stages [1,38].

The current study has its strengths and limitations. A strength of our study is that we followed up all participants individually, hence, could distinguish between apparent reinfections and new infections, whereas previous studies reporting apparent reinfection rates mostly focused on the prevalence before and after treatment, which does not allow to separate new infections from apparent reinfections [39]. A second strength was the inclusion criterion of ≥100 EPG for *T. trichiura* infections at baseline with at least two of the quadruple Kato-Katz slides being positive for *T. trichiura*, hence excluding very light infections, which therefore reduces the diagnostic error. Nonetheless, stool samples were examined by the Kato-Katz technique which is known to have low sensitivity for light STH infections [40,41]. For this reason, the Kato-Katz method may lead to an underestimation of the true prevalence, which in turn could result in an artificial inflation of CRs from undetected residual low-egg count infections post-treatment. Indeed, a comparison of Kato-Katz and qPCR efficacy results 14–21 days post-treatment on Pemba Island revealed significantly lower CRs for ivermectin-albendazole, when two fecal samples before and after treatment were analysed by qPCR [42]. Since the majority of participants had light infections at follow-up, our resulting apparent reinfection rates and new infections should therefore be interpreted with caution. As an effort to overcome this limitation, two fecal samples were collected from different days and two slides were prepared from each sample. Another limitation of our study is that the six months prevalence assessment in Lao PDR overlapped with the national MDA during which most SAC received an additional mebendazole tablet from local health authorities. However, efficacy of mebendazole against *T. trichiura* is known to be low [19,27,43,44], hence, we only expect a minor influence on the prevalence assessment results in this country. Lastly, due to the fact that the short-term efficacy of albendazole alone was low, the sample size for apparent reinfection measurements was limited, hampering meaningful conclusions.

## Conclusion

The results obtained from this study emphasize a strong performance of the ivermectin-albendazole combination at reducing the infection intensity and, consequently, STH-attributable morbidity, while albendazole alone revealed poor long-term outcomes. Our findings support the use of ivermectin-albendazole, a treatment already included in the WHO Model List of Essential Medicines for treating intestinal helminths. However, ivemectin will need to become available at an affordable price to be able to include the drug in STH control programs. Careful decisions on the trade-off between benefits (ivermectin-albendazole *vs* albendazole alone), and costs (several MDA rounds a year), adapted to the epidemiological parasite profile in each setting, have to be made when planning and implementing future control strategies [45–47]. Additionally, as the transmission of *T. trichiura* infections remains high despite effective drugs, future control strategies need to be complemented by improving sanitation and perspicuous information, education and communication (IEC) strategies [1,5,48,49] in order to effectively control and eliminate STH infections more sustainably.

## Supporting information

**S1 Text. Household questionnaire.**
(PDF)

**S1 Table. Short and long-term outcomes and reinfection data for ivermectin-albendazole and albendazole alone against *A. lumbricoides*, hookworm and *S. stercoralis* in Lao PDR and on Pemba Island.**
(PDF)

**S1 Fig. Reported open defecation practices.** Multiple choice question that was asked to one adult household member per screened household.
(PDF)

## Acknowledgments

We would like to express our thanks to all participants for their contribution; the village heads and the local medical teams for their support and commitment; and the team of the Public Health Laboratory–Ivo de Carneri and Lao Tropical and Public Health Institute for all their hard work within this collaboration. We like to thank Jörg Huwyler and Maxim Puchkov for the preparation of the ivermectin placebos.

## Author Contributions

**Conceptualization:** Ladina Keller, Eveline Hürlimann, Jennifer Keiser.

**Data curation:** Ladina Keller, Sophie Welsche, Chandni Patel.

**Formal analysis:** Ladina Keller, Jan Hattendorf.

**Funding acquisition:** Jennifer Keiser.

**Investigation:** Ladina Keller, Sophie Welsche, Chandni Patel.

**Methodology:** Jan Hattendorf.

**Project administration:** Somphou Sayasone, Said M. Ali, Shaali M. Ame, Eveline Hürlimann, Jennifer Keiser.

**Supervision:** Somphou Sayasone, Said M. Ali, Shaali M. Ame, Jan Hattendorf, Eveline Hürlimann, Jennifer Keiser.

**Visualization:** Jan Hattendorf.

**Writing – original draft:** Ladina Keller.

**Writing – review & editing:** Sophie Welsche, Chandni Patel, Somphou Sayasone, Said M. Ali, Shaali M. Ame, Jan Hattendorf, Eveline Hürlimann, Jennifer Keiser.

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
