## [Decision Letter · Decision Letter 0]

3 Jun 2021

Dear Prof. Keiser,

Thank you very much for submitting your manuscript "Long-term effects of ivermectin-albendazole versus albendazole alone against soil-transmitted helminths: results from randomized controlled trials in Lao PDR and Pemba Island, Tanzania" for consideration at PLOS Neglected Tropical Diseases. As with all papers reviewed by the journal, your manuscript was reviewed by members of the editorial board and by several independent reviewers. The reviewers appreciated the attention to an important topic. Based on the reviews, we are likely to accept this manuscript for publication, providing that you modify the manuscript according to the review recommendations. 

Sincerely,

Keke C Fairfax, PhD

Deputy Editor

Keke Fairfax

Deputy Editor

Reviewer's Responses to Questions

**Key Review Criteria Required for Acceptance?**

**Methods**

-Are the objectives of the study clearly articulated with a clear testable hypothesis stated?

-Is the study design appropriate to address the stated objectives?

-Is the population clearly described and appropriate for the hypothesis being tested?

-Is the sample size sufficient to ensure adequate power to address the hypothesis being tested?

-Were correct statistical analysis used to support conclusions?

-Are there concerns about ethical or regulatory requirements being met?

Reviewer #1: hthe methods are adequate and well described.

I have an issue on terminology: see general comment.

Reviewer #2: This manuscript is part of a series of publications describing the design and outcome of a double-blind randomized controlled drug trial performed in three different countries, namely Lao PDR, Tanzania and Côte d’Ivoire. The aim of the overall study is to evaluate the efficacy and safety of ivermectin-albendazole versus albendazole-placebo in Trichuris trichiura-infected individuals. The focus of the current manuscript is on the long term effects, in particular cure rates, egg-reduction rates and reinfection levels, 6 months and 12 month after treatment. Because of the low T. trichuria cure rates at 2-3 weeks in Côte d’Ivoire, the study had to be stopped at that location after the first follow-up time point. This is unfortunate. I would have been interested to see the 6 months and 12 months follow-up in Côte d’Ivoire as well, including the co-infections levels, as well-designed and well-performed longitudinal studies on Soil Transmitted Helminths are rare.

The current manuscript is very clear and I read it with great pleasure. There is some repetition of details with other publications of the same group, but this has the advantage that the manuscript can easily be read without the need to study the other publications. I appreciate the suplementairy documents, including the full study protocol. 

I have some minor comments only.

Line 211-213: concerning quality control of the Kato-Katz slides. Can the authors explain how they dealt time-wise with relabelling and re-examining of the slides in such a way that the detection of hookworm eggs could still be included in the QC? This is also not well explained in the study protocol paper (Patel et al., 2019).

Line 212-215 and further. In Lao PDR, the collected stool samples were also examined for Opisthorchis viverrini and Strongyloides stercoralis. I find it a pity that the authors paid so little attention to the outcomes for these two helminth species. Only a single line concerning the prevalence in Table 1. Will the authors explore further details in an additional manuscript? In particular if molecular diagnosis will be included, this could be an interesting side-topic. Please clarify.

In figure 1 I see (a) marked as S. stercoralis included in the ALB arm. I assume the same was done in the IVM-ALB arm? And all O. viverrini positives were included in the definition of helminths positive? This is not clear from the study protocol paper or the short-term effect manuscript (Hurlimann et al, sumitted). In both previous manuscripts O. viverrini has not been mentioned at all. Some clarification would be appreciated, including some lines in the discussion about both S. stercoralis and O. viverrini.

**Results**

-Does the analysis presented match the analysis plan?

-Are the results clearly and completely presented?

-Are the figures (Tables, Images) of sufficient quality for clarity?

Reviewer #1: the results are well presented and complete

the figures are very nice and clear

Reviewer #2: I very much appreciate figures 2-4. Please indicate that the red dots at baseline and at 6M represent moment of treatment.

**Conclusions**

-Are the conclusions supported by the data presented?

-Are the limitations of analysis clearly described?

-Do the authors discuss how these data can be helpful to advance our understanding of the topic under study?

-Is public health relevance addressed?

Reviewer #1: please see general comments

Reviewer #2: The conclusions are supported by the data.

**Editorial and Data Presentation Modifications?**

Reviewer #1: (No Response)

Reviewer #2: Line 298: an error is indicated for the reference.

**Summary and General Comments**

Reviewer #1: The study is very interesting, it is following, over time, the individual changes in intensity of infection of positive patients treated with albendazole plus ivermectin and albendazole plus placebo.

The article is clearly written and provide very nice figures (# 2, 3 and 4) and therefore in my opinion merit publication.

I have only terminological comments and suggestions for further analysis for the authors to consider.

Main comments

Terminology:

1- In the entire paper the authors mentioned the “long term effect” of the drug. 

a- I do not think this definition is correct because entails that the drug continue to have an “effect” on the parasite after the one observed immediately after initial administration. 

We know that, because the very short half life of the drug used we know this is probably not the case.

The authors concluded on a presence of “long term effect” of albendazole + ivermectin, by comparing the % of individuals at the different classes of infection at the different time points.

In reality if figure 2 is carefully observed it became evident that most of the “ long term “ effect observed by the authors is explained by the much higher “short term effect” of the albendazole + ivermectin.

In other words: if the authors would consider the situation at 3 weeks after treatment as a starting point to measures “long term” effect, they will see that the “additional changes” at 6 months and 1 year are very similar in the 2 arms both in Lao and in Pemba; so my conclusion would be that there are not additional “long term effects” of the drug combination, but that the short term effect is so much better for albendazole + ivermectin, that despite reinfection, (that is the same in the two arms) is still possible to see a big difference in negative individuals after one year from the treatment.

For this reason I suggest not to cite any “long term effect” that does not exist but rather “long term outcome”

I hope the authors will not find this discussion pedantic and will be able to feel the differences.

b- I think is important to clearly explain this issue in the discussion: the combination has not any "long term effect" but the (short term) efficacy is so much better, that the outcomes are still present after one year of treatment, despite reinfection. 

2-I noted that the authors cited frequently “apparent” reinfection, I understand why: they are not sure if the presence of the parasite eggs in the specimen after 6 months is due to a “real” reinfection and not for example due to a time-limited effect of the drug in reducing the egg output of the parasite; in this case the reappearance of the egg in the specimen are would be due not to reinfection but to a restart of the oviposition by the parasite;

In my opinion the results of this study showing that the reinfection is dependent entirely on the site where the study is conducted (and therefore from the differentlevel of environmental contamination) they can conclude that this is a “real” reinfection and not an “apparent” one.

3-line 255 I do not understand very well the formula.

Is the epg calculated on the positive only? Or the negative (epg=0) where included in the calculation of the mean? if this is the case this shouldbe corrected.

4-Line 471 and following. Conclusion, I do not think the authors can extrapolate the impact of the combination therapy in their study to the impact that the combination would have in the context of a control programme. The sample of this study was composed 100% of individuals positive to T. trichiura, (this is very rare occurrence in reality) in case of a much lower prevalence of this parasite in the population, albendazole alone could reduce the prevalence of infections of heavy intensity under the levels indicated by WHO as target for 2030.

In other words this combination therapy is an important additional tools to be used in areas of remaining high prevalence of trichuriasis (that are relatively few) but its application should be carefully considered especially in term of additional cost.

5-Line 494 I do not understand why the authors are surprised by the difference in reinfection between Lao and Pemba: the reinfection depends from the level of contamination of the environment that is dependent not only by the level of sanitation but also from the population density (how many people do contaminate the environment) as they surely know, the population density is much higher in Pemba and as a consequence it is also much higher the environmental contamination and the reinfection rate.

6-Finally I think the authors should mention that the lack of ivermectin at affordable price is an important limitation for the implementation of the combination therapy in control programme 

Minor points

7- Since the data from Cote d’Ivoire are not presented I suggest removing reference to the study conducted in this country are only confusing.

Reviewer #2: This manuscript is very well written, the objectives are clear and relevant for those working in the field of STH control.

In the study protocol paper and the S2-file it is mentioned that stool samples will be examined by qPCR, not only to differentiate the hookworm species, but also to explore the comparison between microscopy and DNA detection procedures. Some PCR data has recently been published in Keller et al., 2020. This publication only describes the Tanzania samples and is limited to the samples collected at baseline and 2-3 weeks after treatment. I would appreciate if the authors could explain in their discussion if we can expect more data to be revealed concerning the qPCR findings of the samples 6 months and 12 months after treatment and also from the two other sites.

PLOS authors have the option to publish the peer review history of their article (what does this mean?). If published, this will include your full peer review and any attached files.

Reviewer #1: Yes: Antonio Montresor

Reviewer #2: No

Figure Files:

Data Requirements:

Reproducibility:

References

---

## [Editor Report · Decision Letter 1]

14 Jun 2021

Dear Prof. Keiser,

We are pleased to inform you that your manuscript 'Long-term outcomes of ivermectin-albendazole versus albendazole alone against soil-transmitted helminths: results from randomized controlled trials in Lao PDR and Pemba Island, Tanzania' has been provisionally accepted for publication in PLOS Neglected Tropical Diseases.

Best regards,

Keke C Fairfax, PhD

Deputy Editor

Keke Fairfax

Deputy Editor

---

## [Editor Report · Acceptance letter]

25 Jun 2021

Dear Prof. Keiser,

We are delighted to inform you that your manuscript, "Long-term outcomes of ivermectin-albendazole versus albendazole alone against soil-transmitted helminths: results from randomized controlled trials in Lao PDR and Pemba Island, Tanzania," has been formally accepted for publication in PLOS Neglected Tropical Diseases.

Best regards,

Shaden Kamhawi

co-Editor-in-Chief

Paul Brindley

co-Editor-in-Chief
